# Neural Payoff Machines: Predicting Fair and Stable Payoff Allocations Among Team Members

**Daphne Cornelisse**[1]    **Thomas Rood**[1]    **Mateusz Malinowski**[3]    **Yoram Bachrach**[3]
**Tal Kachman**[1,2*]

[1]Department of Artificial Intelligence, Radboud University, Netherlands
[2]Donders Institute for Brain, Cognition and Behavior, Radboud University, Netherlands
[3]DeepMind, UK

## Abstract

In many multi-agent settings, participants can form teams to achieve collective outcomes that may far surpass their individual capabilities. Measuring the relative contributions of agents and allocating them shares of the reward that promote long-lasting cooperation are difficult tasks. Cooperative game theory offers solution concepts identifying distribution schemes, such as the Shapley value, that fairly reflect the contribution of individuals to the performance of the team or the Core, which reduces the incentive of agents to abandon their team. Applications of such methods include identifying influential features and sharing the costs of joint ventures or team formation. Unfortunately, using these solutions requires tackling a computational barrier as they are hard to compute, even in restricted settings. In this work, we show how cooperative game-theoretic solutions can be distilled into a learned model by training neural networks to propose fair and stable payoff allocations. We show that our approach creates models that can generalize to games far from the training distribution and can predict solutions for more players than observed during training. An important application of our framework is Explainable AI: our approach can be used to speed-up Shapley value computations on many instances.

## 1 Introduction

The ability of individuals to form teams and collaborate is crucial to their performance in many environments. The success of humans as a species hinges on our capability to cooperate at scale [23]. Similarly, cooperation between learning agents is necessary to achieve high performance in many environments [28, 7, 45, 24] and is a fundamental problem in artificial intelligence [42, 12]. Individual agents are often not incentivized by the joint reward achieved by a team but rather by their share of the spoils. Hence, teams are only likely to be formed when the overall gains obtained by the team are appropriately distributed between its members. However, understanding how collective outcomes arise from subsets of locally interacting parts, or measuring the impact of individuals on the team's performance, remain open problems.

Direct applications exist in multiple domains. One example is identifying the most influential features that drive a model to make a certain prediction [14, 29, 6, 44, 27, 26]; one of the cornerstones of explainable AI [2, 34]. Another example is sharing the costs of data acquisition or a joint venture in a fair way between participants [8, 1], or sharing gains between cooperating agents [20, 41]. In many legislative bodies individual participants have different weights, and passing a decision requires support from a set of participants holding the majority of the weight; different states in the US electoral college have different numbers of electors, and different countries in the EU Council

---

*Correspondence to tal.kachman@donders.ru.nl

of Ministers vary in their voting weight. Here, would like to quantify the true political power held by each participant, or allocate a common budget between them [32, 10].

Cooperative game theory can provide strong theoretical foundations underpinning such applications. The field provides solution concepts that measure the relative impact of individuals on team performance, or the individual rewards agents are entitled to. *Power indices* such as the Shapley value [40] or Banzhaf index [9] attempt to divide the joint reward in a way that is *fair*, and have recently been used to compute feature importance [29]. In contrast, other solutions such as the Core [22] attempt to offer a *stable* allocation of payoffs, where individual agents are incentivised to continue working with their team, rather than attempting to break away from their team in favor of working with other agents. Despite their theoretical appeal, these solution concepts are difficult to apply in practice due to computational constraints. Computing them is typically a hard problem, even in restricted environments [18, 11, 15].

**Our contribution:** We construct models that predict fair or stable payoff allocations among team members, combining solution concepts from cooperative game theory with the predictive power of neural networks. These neural "payoff machines" take in a representation of the performance or reward achievable by different subsets of agents, and output a suggested payoff vector allocating the total reward between the agents. By training the neural networks based on different cooperative solution concepts, the model can be tuned to aim for a fair distribution of the payoffs (the Shapley value or Banzhaf index) or to minimize the incentives of agents to abandon the team (the Least-Core [22, 33, 15]). Figure 1 depicts the two well-studied classes of games on which we evaluate our approach: weighted voting games [32, 10, 11] and feature importance games in explainable AI [14, 29].

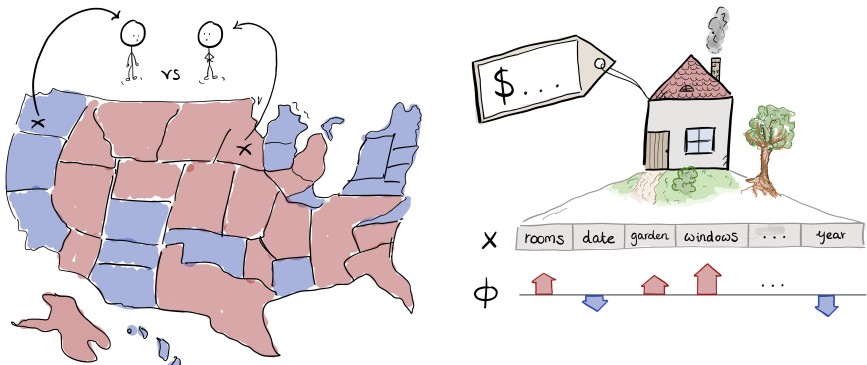

Figure 1: **Evaluation domains for our approach.** *Left*: Weighted voting games (WVGs) model decision making bodies such as the US Electoral College [32, 10]. *Right*: Applying the Shapley value in Feature Importance Games enables quantifying the relative impact of features on the decisions of a model. In this example, a model predicts the price of a house based on several features.

**Weighted voting games (WVGs)** are arguably the most well-studied class of cooperative games. Each agent is endowed with a weight and a team achieves its goal if the sum of the weights of the team members exceeds a certain threshold (quota). We train the neural networks by generating large sets of such games and computing their respective game theoretic solutions. Our empirical evaluation shows that the predictions for the various solutions (the Shapley value, Banzhaf index and Least-Core) accurately reflect the true game theoretic solutions on previously unobserved games. Furthermore, the resulting model can generalize even to games that are very far from the training distribution or with more players than the games in the training set.

**Feature importance games** are a model for quantifying the relative influence of features on the outcome of a machine learning model [14, 29]. Solving these games for the Shapley value (or any other game theoretic measure) provides a way to reverse-engineer the key factors that drove a model to reach a specific decision. This approach is model-agnostic, thus can be applied to make any "blackbox" model more interpretable [2]. One drawback of this approach is the computational complexity of calculating the Shapley value, making such analysis slow even when using approximation algorithms. Our approach provides a way to significantly speedup Explainable AI analyses, particularly for datasets with a large number of instances.

## 2 Preliminaries

We provide a brief overview of cooperative game theory (examined in various books [37, 11]) and briefly discuss how solution concepts in cooperative game theory have been applied in Explainable AI [14, 29, 30, 13, 47].

### 2.1 Cooperative Game Theory

A (transferable utility) **cooperative game** consists of a set $N = \{1, 2, \ldots, n\}$ of agents, or players, and a characteristic function $v : 2^n \to \mathbb{R}$ which maps each team of players, or *coalition* $C \subseteq N$, to a real number. This number indicates the joint reward the players obtained as a team. Games $v$ where $v : 2^n \to \{0, 1\}$ (binary range) are *simple* games.

**Weighted voting games** (WVGs) are a restricted class of simple cooperative games [11], where each agent $i$ has a weight $w_i$ and a team of agents $C \subseteq N$ wins if the sum of the weights of its participants $\sum_{i=1}^{n} w_i$ exceeds a quota $q$. Formally, a WVG is defined as the triple $(\mathbf{w}, q, v)$ with weights $\mathbf{w} = (w_1, w_2, \ldots, w_n) \in \mathbb{R}_{\geq 0}^n$ and quota (threshold) $q \in \mathbb{R}_{\geq 0}$ where for any $C \subseteq N$ we have $v(C) = 1$ if $\sum_{i=1}^{n} w_i \geq q$ and otherwise $v(C) = 0$. If $v(C) = 0$ we say $C$ is a losing coalition, and if $v(C) = 1$ we say it is a winning coalitions. WVGs have been thoroughly investigated as a model of voting bodies, such the US Electoral College or the EU Council of Ministers [32, 10].

The characteristic function defines the joint value of a coalition, but it does not specify how the value should be distributed among the agents. *Solution concepts* attempt to determine an allocation $\mathbf{p} = (p_1, \ldots, p_n)$ of the utility $v(N)$ achieved by the grand coalition of all the agents; an allocation $\mathbf{p}$ is called an *imputation* if for any player $i$ we have $p_i \geq 0$ and $\sum_{i=1}^{n} p_i = v(N)$. [2] This allocation is meant to achieve some desiderata, such as fairly reflecting the contributions of individual agents, or achieving stability in the sense that no subset of agents is incentivized to abandon the team and form a new team. We describe three prominent solution concepts that we use in our analysis.

**The Core.** Rational players may abandon the grand coalition of all the agents if they can increase their individual utility by doing so. The Core is defined as the set of all payoff vectors where no subset of agents can generate more utility, as measured by the characteristic function, than the total payoff they are currently awarded by the payoff vector. As such, the Core is viewed as the set of *stable* payoff allocations. Formally, the core [22] is defined as the set of all imputations $\mathbf{p}$ such that $\sum_{i=1}^{n} p_i = v(N)$ and that $\sum_{i \in C} p_i \geq v(C)$ for any coalition $C \subseteq N$.

**The $\varepsilon$-Core and Least-Core** [33, 15]. Some games have empty cores, meaning that no payoff allocations achieves full stability (i.e. for any imputation $\mathbf{p}$ there exists at least one coalition $C$ such that $v(C) > \sum_{i \in C} p_i$). In such cases, researchers have proposed minimizing the instability. A relaxation of the core is the **$\varepsilon$-core**, consisting of imputations $\mathbf{p}$ where for any value coalition $C$ we have $p(C) \geq v(C) - \varepsilon$. Given an imputation $\mathbf{p}$ the difference $v(C) - \sum_{i \in C} p_i$ is called the *excess* of the coalition, and represents the total improvement in utility the members of $C$ can achieve by abandoning the grand coalition and working on their own. For an imputation in the $\varepsilon$-core, no agent subset $C$ can achieve an addition of $\varepsilon$ in utility over the current total payoff offered to the team (i.e. no coalition has an excess of more than $\varepsilon$). The minimal $\varepsilon$ for which the $\varepsilon$-core is non-empty is called the *Least-Core Value* (LCV). The Least-Core minimizes the incentive of any agent subset to abandon the grand coalition, and the LCV thus represents the degree of instability (excess) under the imputation that best minimizes this instability. We find the set of payoffs associated with the LVC through linear programming (full details in Appendix **??**).

We now discuss two *power indices*, payoff distributions reflecting the true influence a player has on the performance of the team, that fairly allocate the total gains of the teams among the agents in it.

**The Shapley Value** [40] measures the average marginal contribution of each player across all permutations of the players. The Shapley value is the unique solution concept that fulfills several natural fairness axioms [16], and has thus found many applications from estimating feature importance [14, 29] to pruning neural networks [46, 19, 21]. Formally, we denote a permutation of the players by $\pi$, where $\pi$ is a bijective mapping of $N$ to itself, and the set of all such permutations by $\Pi$. By $C_\pi(i)$ we denote all players appearing before $i$ in the permutation $\pi$. The Shapley value

---

[2]In a WVG, the value of a coalition is bounded by 1, so if the grand coalition $N$ indeed has a value of $v(N) = 1$ then a solution would be a payoff vector $\mathbf{p} = (p_1, \ldots, p_n)$ where $\sum_{i=1}^{n} p_i = 1$.

$\phi_i(G)$ of player $i$ is defined as:

$$\phi_i(G) = \frac{1}{N!} \sum_{\pi \in \Pi} [v(C_\pi(i) \cup \{i\}) - v(C_\pi(i))] \tag{2.1}$$

Intuitively, one can consider starting with an empty coalition and adding the players' weights in the order of the permutation; the first player whose addition meets or exceeds the quota is considered the pivotal player in the permutation. The Shapley value then measures the proportion of permutations where a player is the pivotal player.

**The Banzhaf index** [9] is another method for distributing payoffs according to a players' ability to change the outcome of the game, but it reflects slightly different fairness axioms [43]. The Banzhaf index $\beta_i$ of a player $i$ is defined as the marginal contribution of a player across all *subsets* not containing that player:

$$\beta_i(G) = \frac{1}{2^{N-1}} \sum_{C \subseteq N \setminus \{i\}} [v(C \cup \{i\}) - v(C)] \tag{2.2}$$

In practice, we first compute the set of winning coalitions $C^{\text{win}} \subseteq C$ and count, for each player, the number of times it is critical or *pivotal*, that is, $v(\{C\} \setminus i) = 0$.

## 2.2 Speeding Up Explainable AI In Large Datasets

An important application of our work is to speedup Shapley/core computations of many data instances. In machine learning, we train a model $f_\theta$ to learn a mapping from some set of input features to an outcome. For instance, we can train a model to predict the price of a house based on several features, such as the number of rooms, the year it was built, and so on. In such settings, it is desirable, but challenging, to explain the model outputs in terms of the input features. Explainable AI addresses this issue, and recent years showed several applications of game theoretic metrics for measuring feature importance in the machine learning community [44, 2].

The fastest method to approximate Shapley values (also used in the SHAP package) is a Monte-Carlo approach [29]. A number of other methods exist whose runtime and accuracy depend on the number of samples used, usually on the order of several thousands [25, 14, 31, 5]. In a model setting, the characteristic function takes the value of the trained model output for a given instance: $f_{\theta^*}(\mathbf{x})$. The Shapley value of a feature $i$ in a data instance $x$, $\phi_{x,i}$ is the effect that feature has on the model outcome. Sampling based-methods compute the contributions with respect to a base value, which is the average model output across all instances:

$$f_{\theta^*}(x) = \mathbb{E}[f_{\theta^*}(x)] + \sum_{i=1}^{n} \phi_{x,i} \tag{2.3}$$

While effective for obtaining Shapley values of a small set of instances, sampling based methods are not ideal for large datasets because they require a large number of re-evaluation samples per computation. We show how our approach can be employed to speedup Shapley or Core computations of many instances through models that learn representations of feature attribution schemes.

## 3 Methods

Our approach uses machine learning to create game-theoretic estimators. We generate synthetic datasets of games to train our models, spending compute up-front to allow for instant solutions afterwards. Our first domain concerns weighted voting games (WVGs) as they provide a generic framework for studying cooperation in multi-agent settings.

Afterwards, we apply our approach to the Shapley-based feature importance setting. The main idea is that we can speed-up the computations of the relative impact of features on the predictions of a machine learning model. We do this by training models to approximate Shapley values of features, and examine their performance on previously unobserved instances.

### 3.1 Weighted Voting Games

We construct two types of feed-forward neural networks. *Fixed-size* models are trained on games of a specific number of players. The second type of model can take in and predict payoffs for a varying number of players. We refer to these as *variable-size* models. What follows is a formal description of the data and architectures for both models. We now describe our data generation process, models, training procedure, and evaluation metrics used in weighted voting games.

### 3.1.1 Data and Models: Fixed-Size

For each $n$ player game, we generate $G$ independent and identically distributed games. The training dataset is given by $\mathcal{D}_{\text{fixed}}^n = \left\{ \mathbf{X} \in \mathbb{R}^{G \times n}, \mathbf{P} \in \mathbb{R}^{G \times K} \right\}$ where $K$ denotes the number of outputs of interest. Figure 2 depicts the pipeline for generating one data instance. First, we sample a weight vector $\mathbf{w} \sim \text{Beta}(\alpha = 1, \beta = 1)$ on the interval from 1 to $2n$, and a quota $q \sim \mathcal{N}(\mu = \frac{1}{4}(2n+1)n, \sigma^2 = 2n)$. To create games where players are dependent on each other to achieve the task at hand, the quota distribution is parameterized such that the average drawn quota is half of the sum of the players' weights. We then divide the weights by the quota to get the feature vector $\mathbf{x} = \frac{1}{q} \cdot \mathbf{w}$. In other words, we have $n$ features which are the weights normalized as the respective proportion of the quota. Thus, a value $x_i > 1$ indicates that player $i$ is a winning coalition by itself, and needs other players to meet the quota otherwise.

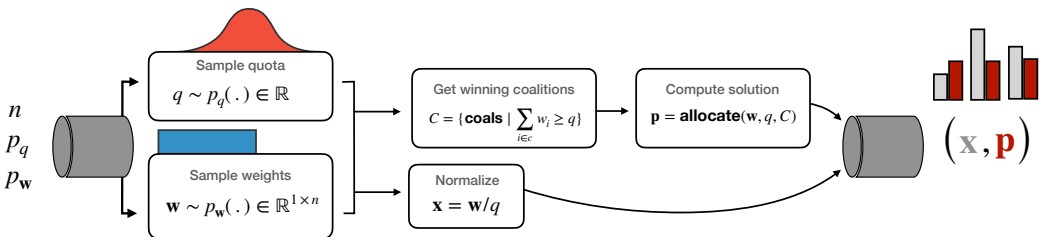

Figure 2: **Procedurally generating and solving one weighted voting game.** We obtain a weighted voting game by sampling $n$ weights and a quota. The weights are divided by the quota to create the inputs $\mathbf{x}$. We solve the game for each considered solution concept to create our targets $\mathbf{p}$.

We train models to predict the three solutions: the Least-Core, the Shapley values, and the Banzhaf indices. For the Shapey values and the Banzhaf indices, the model predicts the payoff allocation $(p_1, \ldots, p_n)$ so $K \equiv n$. For the Least-Core solution, we train the model to not only predict the payoff allocation $p_1, \ldots, p_n$, but also to predict the least core Value $\varepsilon_{min}$ (so in this case the model has $K = n + 1$ outputs). For each $n$ player game, we produce a model $f_{\boldsymbol{\theta}} : \mathbb{R}^n \to \mathbb{R}^K$, where $\boldsymbol{\theta}$ are the model parameters. We use deep feedforward networks for $f_{\boldsymbol{\theta}}$. Appendix **??** contains the full details about the experimental setup.

### 3.1.2 Data and Models: Variable-size

For the variable-size case, we consider a maximal number of possible players $M$ and pad the inputs with zeros for games with less than $M$ players. Hence, we generate a single dataset $\mathcal{D}_{\text{var}} = \left\{ \mathbf{X} \in \mathbb{R}^{G \times M}, \mathbf{P} \in \mathbb{R}^{G \times K} \right\}$. Similarly to the fixed-size dataset, the feature matrix $\mathbf{X}$ contains the normalized weights and $\mathbf{P}$ the corresponding solutions with either $K = M + 1$ or $K = M$, with the ground truth output vector again padded with zeros when there are fewer than $M$ players. Hence, we allow for the prediction of up to $M$ players, and we shuffle the data so that players are located at random positions. The least core Value (LCV) is not shuffled but stored at the last element of each row.

We produce a single model $g_{\boldsymbol{\eta}} : \mathbb{R}^n \to \mathbb{R}^K$ that can be used for different number of players $n$, where $\boldsymbol{\eta}$ are the model parameters. The model learns to allocate the joint payoff among at most $M$ players. During prediction time we pad the input with zeros when there are fewer than $M$ players, and we redistribute the payoffs allocated to non-player entries among the players according to their original share of the joint payoff. Figure 3 graphically depicts the padding procedure and basic model architecture.

### 3.1.3 Training and Evaluation

During training we minimize the Mean square error (MSE) between the true and predicted solutions. For the variable-size models we also include the $M - n$ padded locations so that the model learns not to allocate value to non-player entries.

**Evaluation metrics.** Given a predicted payoff vector $\hat{\mathbf{p}} = (\hat{p}_1, \hat{p}_2, \ldots, \hat{p}_n)$, we consider multiple metrics for assessing the models' performance. First, we quantify the models' predictive performance via the Mean absolute error (MAE), defined for each game as $\text{MAE} = \frac{1}{n} \sum_{i=1}^{n} |p_i - \hat{p}_i|$, where $n$ is the number of players. For the Least-Core there is another natural game theoretic metric.

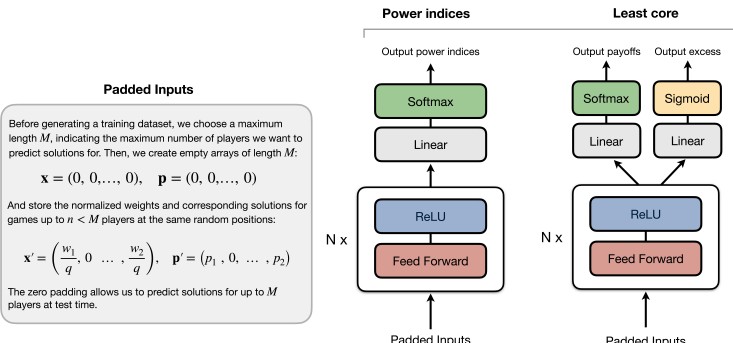

Figure 3: **Neural architectures for the variable-size game predictions.** We add zero-padding to allow one model to predict solutions of games of variable sizes. We choose a maximum length $M = 20$, and add zero-padding to games with less than $M$ players. Full details on our data in Section 3.1.2

The goal of the Least-Core is to minimize the incentives of any subset to abandon the current team and form its own sub-team. Given a suggested payoff vector $\hat{\mathbf{p}}$, the maximal excess $v(C) - \sum_{i \in C} \hat{p}_i$ over all possible coalitions $C$ measures the incentive to abandon the team, and serves as a good measure for the quality of the model.

**Test data.** We sample weights $\mathbf{w} \sim \text{Beta}(\alpha, \beta)$ with varying parameters for $\alpha$ and $\beta$ to assess our models' ability to generalize to previously unobserved instances their ability generalize to games far outside of the training distribution (full details in Table **??**, Figure **??**).

## 3.2 Feature Importance Games

We perform an experiment to show that neural networks can provide a faster alternative for measuring feature importance at scale. We select a dataset (details in Appendix **??**), train a model, and use those to construct a dataset from features to Shapley values using the SHAP KernelExplainer [29] Following, we partition our dataset into a train and test, and incrementally change the proportions between the two. For each increment, we train a model for 100 epochs and test it on the remainder of the unseen instances.

# 4 Results

We present experimental results that allow us to assess how well neural models are able to learn a representation of the various solution concepts. We first describe the predictive performance of neural networks in the WVG setting, and consider properties of these solutions that make them hard to learn. We then consider the explainable AI domain, and study the performance and sample complexity of Shapley feature importance prediction.

## 4.1 Weighted Voting Games: Evaluation

For our WVG analysis we train a selection of fixed-size neural networks $f_{\boldsymbol{\theta}^*}^N$ for each number of players $n \in \{4, 5, \ldots, 19, 20\}^3$ on $G = 5,000$ games each. We also train a single variable-size model $g_{\boldsymbol{\eta}^*}$ that is trained on one dataset containing $G = 17,500$ games in total, consisting of $2,500$ games for each number of players $n \in \{4, 5, \ldots, 9, 10\}$ games. The data is padded with zeros to allow for payoff allocation up to $M = 20$ players (see details in Section 3.1).

### 4.1.1 Predictive performance

Table 1 shows that the ability to handle games of variable numbers of players comes at the cost of having a lower accuracy. However, even for variable-size models, and even under a significant distribution shift, the errors in predicting all solution concepts are low. We further note that the error in predicting Least-Core based payoffs are generally larger than for the Shapley and Banzhaf

---

[3] A direct computation of the Shapley value requires enumerating through a large list of permutations, which becomes computationally very costly when there are many players. Hence, for games with $n = 9$ players or more, we use Monte-Carlo approximations for the Shapley value to obtain the ground-truth solution. See full details in Appendix **??**)

power indices. One possible reason is that for these solutions, there are many cases where a small perturbation in the weights or quota results in a large perturbation of the Least-Core solution [17, 48].

Table 1: Comparison of predictive performance across test datasets and solution concepts.

| | least core payoffs | | least core excess | | Shapley values | | Banzhaf indices | |
|---|---|---|---|---|---|---|---|---|
| Dataset | Mean MAE | | MAE | | Mean MAE | | Mean MAE | |
| | Fixed | Variable | Fixed | Variable | Fixed | Variable | Fixed | Variable |
| In-sample | 0.030 | 0.043 | 0.015 | 0.034 | 0.019 | 0.022 | 0.018 | 0.028 |
| Out-of-sample | 0.030 | 0.044 | 0.015 | 0.027 | 0.018 | 0.036 | 0.018 | 0.056 |
| Slightly out-of-distribution | 0.029 | 0.028 | 0.015 | 0.050 | 0.018 | 0.019 | 0.017 | 0.018 |
| Moderately out-of-distribution | 0.030 | 0.035 | 0.014 | 0.029 | 0.018 | 0.026 | 0.018 | 0.032 |
| Significantly out-of-distribution | 0.031 | 0.045 | 0.014 | 0.036 | 0.018 | 0.029 | 0.018 | 0.039 |

We now highlight several conclusions from our analysis of the results.

**Fixed-size models display stable performance across solution concepts.** Our first observation is that the fixed-size neural networks are adept at estimating solutions across the three considered concepts. They outperform standard multinomial regression models by 45 to 95 % across solution concepts (full details Appendix **??**). The average error per player ranges from 0.005 to 0.087 (least core), 0.004 to 0.069 (Shapley values) and 0.012 to 0.067 (Banzhaf indices). Table **??** provides a complete summary of our models' in-sample performance.

**Fixed-size models are robust to shifts in the weight distribution.** As shown in Figure 4, the predictive performance (Mean MAE) of the fixed-size models is consistent across test datasets. To account for the natural decrease in the MAE as $N$ increases, we also display the average payoff per player $(1/n)$. As expected, the error scales approximately with the average payoff per player.

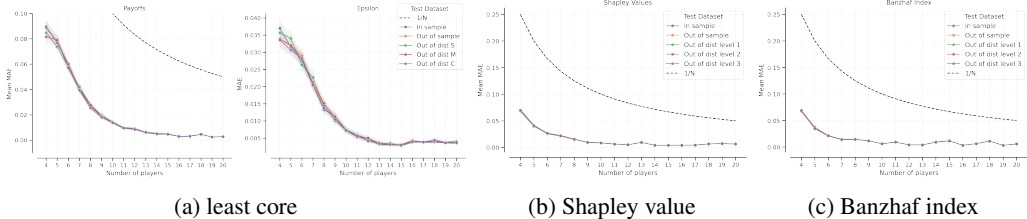

| (a) least core | (b) Shapley value | (c) Banzhaf index |
|---|---|---|

Figure 4: **Performance fixed-size models.** We evaluate the performance on five test sets with 1000 samples per $n$-player game. Figures show the MAE and 95 % confidence intervals for each solution concept. The black dashed line indicates the average payoff per player as a function of $n$.

**Variable-size models are robust to shifts in the weight distribution.** Figure 5 shows that the variable-size models are also able to generalize outside the training distribution. Across test sets, we observe a stable performance that decays with the number of players, as is expected. For the least core, we see that there is an abrupt decrease in performance for the excess beyond 8 players.

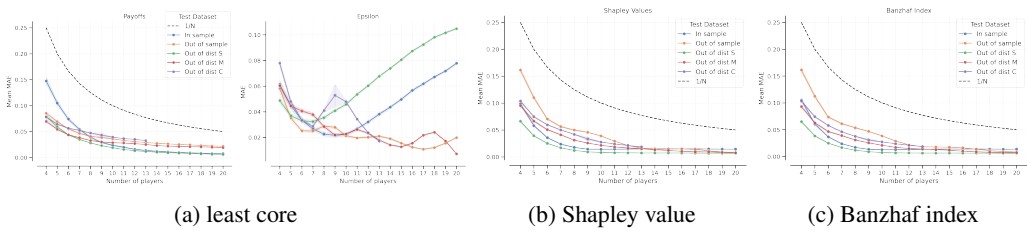

| (a) least core | (b) Shapley value | (c) Banzhaf index |
|---|---|---|

Figure 5: **Performance variable-size models.** We evaluate the performance on five test sets with 1000 samples per $n$-player game. Figures show the MAE and 95 % confidence intervals for each solution concept. The black dashed line indicates the average payoff per player as a function of $n$.

**Variable-size models generalize to a larger number of players.** Our main objective is to investigate how to leverage machine learning to perform scalable value estimation in large multi-agent settings. To test this, we train the variable-size models on games up $n = 10$ players, and evaluate on $n+1, n+2, \ldots, n+10$ players (full details in section 3.1.2). Figure 5 demonstrates that there is no significant decrease in performance for games with more than ten players across solution concepts: variable-size models are able to extrapolate beyond the number of players seen during training. This suggests that there is be valuable information in the small player games such that games of larger sizes can be inferred.

**Fixed-size models outperform variable-size models.** Table 1 contains the Mean MAEs across solution concepts and test sets. Corresponding error distributions are showed in Figure 6. From these, we conclude that in almost all settings the fixed-size networks outperform the variable-size networks. Variable-sized networks tend to have larger errors for all predicted variables and display more variance in their predictions.

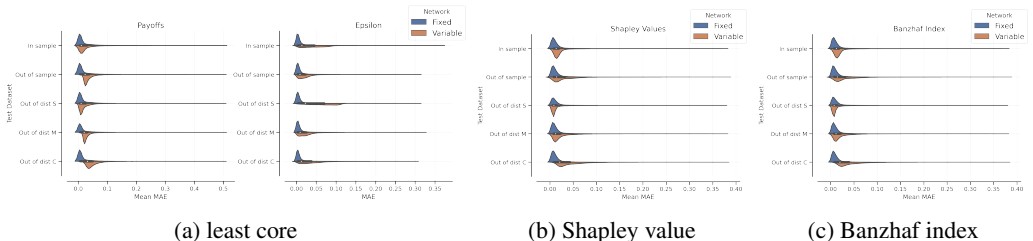

(a) least core          (b) Shapley value          (c) Banzhaf index

Figure 6: **Comparing overall performance fixed-size vs. variable-size networks.**

### 4.1.2 Discontinuities in the Solution Space

The function mapping from game parameters to solutions contains discontinuities. Discontinuous jumps emerge from the players' interdependence and the effect of the quota and are difficult for a model to learn. We analyze two examples that demonstrate how our models respond to such transitions.

**Solution concepts are step-wise constant functions which are difficult to capture.** Consider taking a WVG and changing the weight of a player. This only changes the game theoretic solutions when it restructures the subset of winning coalitions. Thus, the function outputs the same value until the weight reaches a certain threshold where the structures of winning coalitions change, at which point the solution can change drastically. Learning these kind of functions is difficult, as the error around the discontinuity point is often large.

Our analysis examines an $n$ player game with a fixed weight vector $\mathbf{w} = (w_1, w_2, \ldots, w_n)$ and considers an array of quotas $\mathbf{q} = (q_1, q_2, \ldots, q_K) = (\min(\mathbf{w}), \min(\mathbf{w}) + \epsilon, \ldots, \sum_{i=1}^{n} w_i)$, where $\epsilon = 0.1$. We solve the game for each combination of the fixed weights and the changing quota in $\mathbf{q}$ to obtain a matrix $\mathbf{P}$. Figure 7 shows the fixed-size model predictions for two selected WVGs. The models capture the overall effect of the changes to the quota, but do poorly close to the discontinuity point (where the ground truth solution incurs a large change).

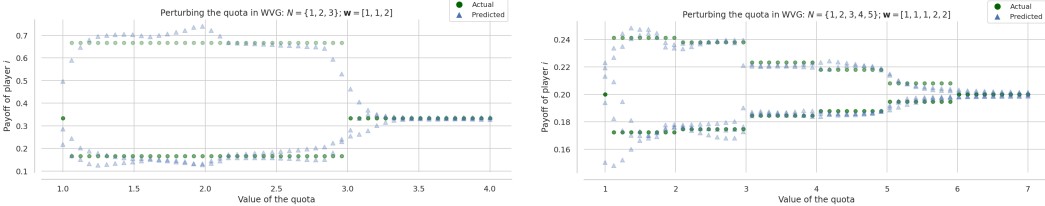

Figure 7: **Capturing step-wise jumps.** Individual actual payoffs (green dots) together with the model predictions (blue triangles) as the value of quota increases by $\varepsilon = 0.1$ increments (full description in Appendix **??**).

## 4.2 Application: Model-based Approach to Explainable AI

**Models speedup XAI analyses by more than 8 times.** Various domains involve datasets with a large number of features, sometimes containing thousands or millions of data instances. A prominent example is language modelling, where large bodies of text are used to perform machine translation, or generate new text [38]. Here, explaining features in terms of the model outcome for every data instance, for example by means of the Shapley value, takes an extremely long time.

Our approach is targeted at speeding up feature importance computations of many instances. We perform a simple experiment to demonstrate the effectiveness of a model-based approach for explaining model decisions. We take the Melbourne Housing dataset [39] and obtain 9000 instances by 13 features after preprocessing steps (encoding the categorical features and standardization). First, we generate a ground-truth dataset by setting the number of samples in the SHAP package (KernelExplainer) to 5000 samples. The number of samples in the KernelExplainer determines the number of times a model is re-evaluated for every prediction. A higher number of samples will result in more accurate (and lower variance in the) Shapley values, but require a longer compute time. Computing the Shapley values on our 9000 samples takes 4.13 hours. Figure 8 (left) shows that the inference time scales approximately linearly with the number of re-evaluation samples.

To contrast the difference in computation time, we compute Shapley values on 10 % of the data (computation time is 29 minutes) train the neural network in 30 seconds, and predict Shapley values on the remaining 90% of the data (less than a second). All together, this procedure takes roughly 12 % of the time of it took to compute Shapley values for the entire dataset. This speeds-up the whole procedure by 8x while keeping a reasonable prediction quality on the remaining unseen data (Figure 8 right).

Next, we use SHAP again to compute Shapley values for 90% of the data, 8100 instances, this time with the default number of model re-evaluations samples, yielding slightly accurate Shapley values (the default is $2 \times$ number of features $+ 2048$). We quantify the trade-off in prediction quality as the by comparing the model predicted Shapley values to the Shapley values obtained with the default number of samples.

Concretely, we measure the Mean Mean Absolute Error (MMSE) of our model predictions and the SHAP package to the ground-truth dataset. We note that speedup does come at a slightly reduced empirical performance: the model MSE on the test set is, MSE $= 1.15 \times 10^{-3}$, compared to the MSE with SHAP: MSE $= 7.4 \times 10^{-5}$. However, we expect that hyperparameter tuning and better preprocessing will improve the quality of the model predictions.

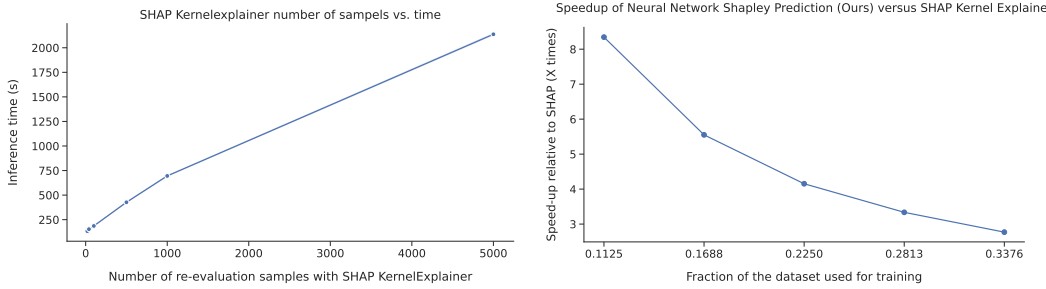

Figure 8: **Speeding up Shapley value computations.** *Left*: The inference time, here the time it takes to compute the Shapley values of 9000 instances, scales approximately linearly with the number of samples. *Right*: The x times speedup provided by our model-based approach contrasted by the fraction of data used for training.

**Neural models are sample efficient.** We also examine the sample efficiency and performance of neural networks trained to predict Shapley values on sets of features. Here, we consider two well-known datasets: the classical UCI Bank Marketing dataset [35] (17 features, 11,162 observations) and the Melbourne Housing dataset [39] (13 features, 34,857 observations). For the full details, see Appendix **??**). Overall, we find that neural networks achieve high performance on the test set with very few training samples. Figure 9 displays the models' performance (in RMSE) as a function of samples available for training. We observe that the error decays approximately exponentially with the proportion of data used for training on both the Banking and Melbourne datasets. A model

trained on just 1 percent of the Melbourne dataset has an RMSE of 19.72 on the resultant test set. Increasing the number of training samples to 3 percent results in a RMSE of less than 0.07, a 99.6 percent decrease in error. We observe a similar trend for the Banking dataset: the RMSE decreases with 93.8 percent (from 4.10 to 0.25) when the available training data increases from 0.5 to 4 percent. These results highlight the strength of this approach: the computational cost of training the Shapley prediction network is very small as compared to the speedup obtained on the vast majority of the data (and any subsequent instances analyzed later).

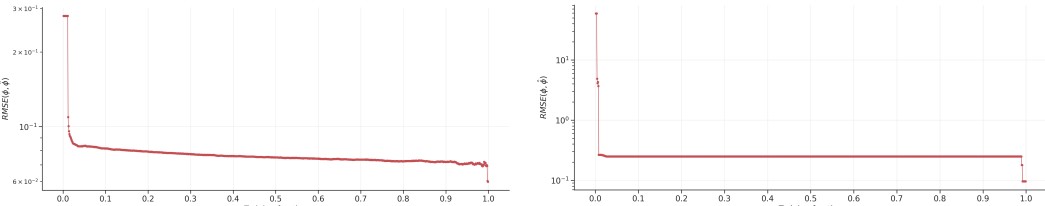

Figure 9: **Root Mean Squared Error as a function of training fraction.** The RMSE between the actual and predicted Shapley values scales approximately as a power law with the training fraction. *Left*: UCI Banking dataset *Right*: Melbourne Housing dataset.

## 5 Discussion and Conclusion

We considered "neural payoff machines", models that take in a representation of a cooperative game and solve it to produce a distribution of the joint gains of a team amongst its members. Our analysis focused on two concise representations of a characteristic function: Weighted voting games, and feature importance games for explainable AI. Our analysis shows that neural models capture cooperative game solutions well, and can generalize well outside the training distribution, even extrapolating to more players than in the training set. However, we observed that the least core is a harder solution concept to learn than the Shapley value and Banzhaf index. Potentially, the fact that the least core is complex and can only be computed by solving multiple Linear Programs makes a hard concept to grasp from examples.

We showed that, with a dataset of 9000 instances, our model-based approach to explainable AI lead to a speed up of more than 8x compared a sampling based approach (SHAP, based on the LIME algorithm). While this comes with a small decrease in prediction quality, we believe that more suitable training procedures can lead to models with performance on par with SHAP. It is easy to see how this win in computation translates to larger datasets. Consider a similar dataset with 100,000 or a million entries. Computing Shapley values on the entire dataset with SHAP will take weeks (40 or 400 hours), while the time for our procedure remains relatively the same: building the train set and training the neural network only takes a fraction of the time required by the sampling-based approach: the analysis using our method would take a few hours.

The methods we discussed can drive better analysis of decision making bodies and for approximating feature importance, but such methods can also be applied to other classes of cooperative games, so long as one can generate data consisting of game samples and their solutions. For instance, the same approach can be used for other cooperative representations, such as graph-based games [15, 3, 4] or set-based representations [36]. The direct applications that we have considered in this paper are also intrinsically valuable on their own, as we allow speeding up explainable AI analysis and political influence estimation. Some questions remain open for future work. Can similar techniques be used for non-cooperative games or for other types of cooperative games? Can this approach be applied to other solutions such as the Kernel or Nucleolous [11]? Finally, are there better neural network designs, such as graph neural networks or transformers, that can exploit invariances or equivariances in the game's representation or be better equipped to deal with sharp discontinuities in the solutions?

### Acknowledgements

This work was partly supported for T.K, D.C and T.R by ELLIS- the European Laboratory for Learning and Intelligent Systems, the Ethereum foundation, Lineage logistics research grant and Kells Establishment.

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
