# OpenReview forum: "Neural Payoff Machines: Predicting Fair and Stable Payoff Allocations Among Team Members"
_NeurIPS.cc/2022/Conference — NeurIPS 2022 Accept_

### Official Review · Reviewer_6UQP · 2022-07-11

**Rating:** 4
**Confidence:** 4
**Soundness:** 3 good
**Presentation:** 3 good
**Contribution:** 2 fair

**Summary:**

This paper applies a multi-layer perceptron to learning to approximate the Shapley value of a weighted voting game (where the input is a description of the voting weights of each player, and the output is a vector of Shapley values for each voter).  The fixed-dimension case (for which the number of voters is constant) and variable dimension case are both considered; the variable dimension case is handled simply by zero-padding (i.e., non-players are added with weight 0 to every empty slot).

Although weighted voting games are simple, approximating their Shapley value within a multiplicative factor is known to be intractable in the worst case.


**Questions:**

Do you have performance numbers (both timing and prediction quality) for any scenario where the use of something like SHAP would be intractable?

**Limitations:**

see strengths/weaknesses

**Strengths And Weaknesses:**

The basic idea is sound: we sometimes want the Shapley value (to estimate the relative influence of different input features on the output of an ML model, or to estimate the relative power of different voters).  However, it's intractable to compute exactly.  So let's try approximating it by training a neural network on small instances.  It seems to work pretty well on the instances that they test on.

One of the main motivations for this work is scalability.  It can be exponentially expensive to compute the Shapley value as the number of participants increases, making an approximation approach valuable.  Clearly the network with a fixed number of participants won't be much use in large-scale situations, since the training data needs to be as large as the test data.  On the other hand, the paper shows that you can get decent performance by training on a dataset with about half as many participants as in the test set (using the zero-padding variable-dimension approach).  This is of course better than nothing, but I'm skeptical that an approximate doubling of the feasible size of input is going to be sufficient in settings that actually need scale.

The paper is clearly written, with the exception of the material on explainable AI.  This domain is an important part of the paper's motivation, but its treatment feels like something of an afterthought.   I can take a guess at what is going on here (the "value" created by a given agent is going to be either the average change in a binary classification, or the average change in regression output, averaged over all orderings), but it would be nice if this were somehow made explicit.  Section 3.2 doesn't really give any details about the actual procedure (what is a "sampled sub-dataset"?  Are you sampling rows (instances) or columns (features)?  Appendix D.2 doesn't refer to this at all.).  Section 2.2 is similarly vague.

Overall, I don't think this work clears the significance bar for publication at NeurIPS.  It proposes a standard feedforward architecture for prediction in a specific scenario.  The scenario is interesting but not really deeply explored.  There is no explicit comparison to baselines, so it's hard to tell what we actually gain.  In particular, we don't see any performance numbers at all.  How large is the performance difference vs standard techniques?  Are you able to do well on instances that take SHAP weeks, for example?

__minor comments__

- p.5: "we redistribute the payoffs allocated to non-player entries": How necessary does this turn out to be?  (I.e., is a substantial fraction of payoff often allocated to non-players?)

- p.6: The first paragraph of s.3.2 is very repetitive

- On a related note, it's not clear from the main text what the ground truth for figure 7 is meant to be.

- Figure 2 is unreadably small, and also seems to make the PDF very slow to render.

---

> ### Author Response · Authors · 2022-08-02
> **Response to reviewer 6UQP**
>
> Thank you for the thorough review and helpful comments. Indeed, our method is based on training neural networks to predict solution concepts for cooperative games. Before addressing your remarks, we would like to emphasize that our work considers two other prominent game-theoretic solution concepts aside from the Shapley value; the Banzhaf index, which captures different fairness axioms than those characterizing the Shapley value, and the Core, which ensures the stability of teams (the cooperation of members of the coalition). While the Shapley value is indeed the most widely used interpretability metric, other metrics provide distinct, but equally interesting ways to acquire an intuitive understanding of model predictions (see also our responses to reviewer qYgj).
>
> **We have cleaned up the presentation regarding Explainable AI (XAI), which focuses on the Shapley value (and do a better job in pointing the reader to the literature that discusses these procedures in detail)**. Shapley XAI analysis takes a trained model and examines a specific input instance (“row”), and then outputs the relative impact the features ("columns") had on the prediction of that specific instance. These are indeed calculated through a sensitivity analysis: the SHAP package takes many samples of “perturbed instances”, where in each such perturbed instance a subset of features are replaced with their dataset average, while the remaining features retain their value in the analyzed instance; The analyzed model is called on the perturbed instance to generate a counterfactual output. By comparing the average model output for perturbed instances where feature f had its original value, and contrasting it with the average model output where feature f was replaced with the dataset mean value for that feature, we get an estimate that feature f has on the prediction.
>
> **Our goal is not to replace SHAP, but rather to achieve a speedup when applying it to many instances. As per our comments to reviewer qYgj, we achieve the speedup by training a neural network to approximate (“distill”) the Shapley value computation**.
>
> We now more fully explain the procedure and analyze the achieved speedup versus the SHAP baseline in **Appendix I**. We take a small fraction of the original dataset (“train subset”), and use the SHAP package to compute Shapley values of features on this dataset. Based on the train subset, we train a neural network to take feature values and output their Shapley values. We then apply this trained neural network rather than the SHAP package to the remainder of the dataset. When doing so, most of the time is spent in building the train subset, i.e. applying the SHAP method to the small fraction of the original dataset. Training the neural network is quick, and applying the trained network to the remaining instances is very quick. We show that even for our relatively small dataset of ~9000 instances, our procedure takes 29 minutes rather than 4.13 hours (an 8x speedup). Now, consider a similar dataset with 100,000 or a million entries. SHAP would take weeks (40 or 400 hours), but the time for our procedure hardly changes: building the train set and training the neural network would take the same amount of time, and as each instance is computed in fractions of a second, the analysis using our method would take a few hours. This comes at the cost of a slightly reduced predictive performance, but our empirical evaluation indicates the error is still very small (see **Appendix I**, MSE < 1.15e-3).
>
> We’d also like to highlight that while our neural network architecture is indeed a simple feedforward network, we apply a novel data augmentation pipeline to allow training of the variable-sized models. We create synthetic instances placing the true entries in random locations (and padding the rest) and show that this is sufficient to train a model that generalizes to games with more players than it had ever seen in the train set. We’ve emphasized this in the updated version of the paper.
>
> You expressed doubts about our models generalizing to games with far more players. However, note that even as is, **our framework already provides value in estimating the least-core in larger games than current methods can handle**. The LP of the core has $2^n$ constraints, and there are no existing methods for effectively approximating the least-core (as do exist for Shapley). Hence, our current results already allow processing somewhat larger games (e.g. the least core is computable within reasonable time for 10 players, and we make it tractable to games of 25 players see **Appendix L** for an example and discussion.
>
> As for the redistribution step, it is required in the majority of cases, but the amount redistributed is very small. The fraction allocated to non-player entries is $6.6$ x $10^{-7}$ for the Shapley values, $7.0$ x $10^{-4}$ for the Banzhaf index; 0.032 for the least core. We have of course fixed the rest of the minor points - thank you!

---

> > ### Author Response · Authors · 2022-08-09
> > **Thank you for all your comments**
> >
> > Thank you for all your comments and suggestions, we have taken and implemented your feedback which has made the paper much better.
> >
> > We would be happy to discuss our corrections or any other feedback you have that we did not address as we are nearing the end of the rebuttal period.

---

### Official Review · Reviewer_JZrJ · 2022-07-11

**Rating:** 7
**Confidence:** 4
**Soundness:** 4 excellent
**Presentation:** 4 excellent
**Contribution:** 4 excellent

**Summary:**

The paper discusses using neural computation to compute values of cooperative game theory in weighted voting games and feature importance games.  For weighted voting games, models for both fixed and variable numbers of players are given.  While fixed models outperformed variable models, results on random games indicates that both models do relatively well.  Furthermore, predictions fro feature importance games are quite good after training on only a small amount of training data.

**Questions:**

See comments in the previous section.

**Limitations:**

I think further exploration of failure cases would be appropriate at some point

**Strengths And Weaknesses:**

Strengths:

+ The paper does a good job of describing and introducing the problem.

+ The presentation of the paper is clear and clean.

+ This paper implements and tests a good, simple, and useful idea: Can neural computation to computer values of cooperative game theory.

+ The analysis of the results is good and detailed.

Weaknesses:

- It would be fun to illustrate how well the trained neural networks perform on known problems rather than on just random games.

- (Table 1) Suggestion: I think it would be good to understand how significant the failures are when the neural network fails (e.g., significantly out of distribution).  Is there any way to detect when it would fail?

A few minor nitpicks:

- It’s probably right in front of my eyes, but I could see where p(C) was defined on line 111

- Line 260: fix “at a the cost” — there’s clearly an extra word

---

> ### Author Response · Authors · 2022-08-02
> **Response to reviewer JZrJ**
>
> We are pleased you appreciate our work and thank you for the very valuable remarks!
>
> You suggested evaluating our models on known real-world problems. This is a great idea, so we have used our models to predict Shapley values, the Banzhaf index, and the Least core to predict the voting power of member states in the European Council; This allows estimating the true influence of countries on the EU’s decision making [1] (the actual power a country has can be much higher than its proportion of total votes). Overall, we find that our models are very close to the ground-truth solutions (Mean MAE of 0.0025 for Shapley, and Banzhaf, 0.017 for the Core). Also, our models are adept at capturing the discontinuous power changes resulting from perturbations to the member states’ weights or quota. See the new **Appendix L** for the full analyses and figures.
>
> There exist a number of other real-world use cases of our work aside from measuring the voting power of members in the European Council. For example, solution concepts from cooperative game theory have been used in research on stable matching mechanisms in college admission markets [2], efficient water allocation or sharing costs from water resources [3, 4], sharing the costs of social ridesharing [5], and more. We briefly elaborate on applications in the new **Appendix M**.
>
> We have addressed the typos in our main paper. $p(C)$ is changed to $\sum_{i=1}^C p_i$ -- the sum of the payoffs in a particular coalition $C$.
>
> You’ve asked regarding possible limitations or failure cases of our models. Even though our models are fast and robust to shifts in the weight distributions (i.e. they do well outside the training distribution), a qualitative analysis revealed several interesting limitations. We highlight failure cases that concern the least core, a solution concept with two properties not shared by Banzhaf or Shapley. First of all, the least core is defined as the set of feasible payoff vectors associated with the smallest excess value (definition in line 211). One challenge arises when there are a number of correct solutions in the least core, which is not a rare phenomenon. This can be particularly problematic in cases where more than one player is required to form a winning coalition, and our Linear Program (a Simplex-based method) returns a corner solution. For example, it will allocate the full joint payoff to one arbitrary player in the winning coalition, while giving the others nothing. Such games yield among the largest errors in our test dataset. An open question is what set-based loss objectives are effective for training models on data with multiple correct solutions [6].
>
> The second form of failure concerns feasibility. Solving for the least core of a weighted voting game (WVG) is done through a Linear Program (details in **Appendix C**) under several hard constraints. Specifically, a solution is only feasible when, for each winning coalition $C \in C^{win}$, we have that: $v(C) = \sum_{i=1}^C p_i \geq 1 - \varepsilon$. We find that only a small fraction of the model predicted solutions are feasible, that is, meet these hard constraints. Specifically, the percentage of infeasible solutions ranges between 50 and 5.5 % for games with four to seven players and is between 10 and 0.2 % for games with more than eight players. This is not surprising, however, as we did not include a penalty term for predicting infeasible solutions during training. Both of these cases provide interesting challenges for future work. We wrote up a full analysis of hard cases in **Appendix K**.
>
> We hope to have addressed the reviewer's comments in sufficient detail. Once again, we warmly thank the reviewer for the useful and fun comments!
>
> ---
>
> [1] Leech, D., Designing the voting system for the Council of the European Union. Public Choice, (2002)
>
> [2] Biro et al., The Large Core of College Admission Markets: Theory and Evidence. In ACM EC (2022)
>
> [3] Allocating costs or rewards from water resources: Dinar, Ariel, Aharon Ratner, and Dan Yaron. "Evaluating cooperative game theory in water resources." Theory and Decision 32.1 (1992): 1-20.
>
> [4] Mirzaei-Nodoushan, F., Bozorg-Haddad, O., & Loáiciga, H. A. (2022). Evaluation of cooperative and non-cooperative game theoretic approaches for water allocation of transboundary rivers. Scientific Reports, 12(1), 1-11.
>
> [5] Allocating costs of social ridesharing: Bistaffa, Filippo, et al. "A cooperative game-theoretic approach to the social ridesharing problem." Artificial Intelligence 246 (2017): 86-117.
>
> [6] Zaheeret al. Deep sets. NIPS (2017)

---

> > ### Comment · Reviewer_JZrJ · 2022-08-08
> > **thanks**
> >
> > Thanks for the response and modifications.  I believe they further strengthen the paper.

---

> > > ### Author Response · Authors · 2022-08-08
> > > **Thank you**
> > >
> > > We thank you for the feedback and all your useful comments, it really helped shape the paper and make it much, much better!

---

### Official Review · Reviewer_qYgj · 2022-07-11

**Rating:** 5
**Confidence:** 3
**Soundness:** 3 good
**Presentation:** 3 good
**Contribution:** 2 fair

**Summary:**

The authors propose a heuristic, deep learning approach to computing agent/feature importance in complex cooperative game theoretic settings. Such setting are known to be hard to solve and understand exactly (for example computing the Shapley value in voting games is #P-complete), motivating the study of the paper.

**Questions:**

- Can the authors compare their results to previous work/non deep learning based results? Does the deep learning approach lead to significant improvement which warrants the downsides of having to use a black-box, hard to understand algorithm?

**Limitations:**

In terms of societal impact, I think it is nice that the authors have applications to interpretability. Limitations are discussed in Section 4.1.2.

**Strengths And Weaknesses:**

- The authors’ algorithms seem to perform well according to the experiments
- The authors’ proposed framework has several proposed and illustrated applications, including solving voting games and understanding feature importance in hard-to-understand models
- The authors look at several metrics of interest for measure contribution/importance, including shapley values, Banzhaf indexes, and cores. The flexibility provided by the framework seems nice and useful.

Weaknesses:
- My main question and “issue” with the paper is that I do not get a good sense here of whether the neural net approach is actually necessary, or if it is actually overkill. For example, if the goal were just to compute Shapley values fast, approaches based on sampling are known and seem to be relatively efficient. How does the current work compare to those?
- Maybe things are more complicated when it comes to compute the actual outcomes rather than just computing shapley values, but this needs to be discussed more. There is indeed motivation for a heuristic approach in that these problems are hard to solve exactly, but it seems the authors jumped to a neural net before exploring simpler heuristics on very structured problems/it is not clear to me why this brings compared to possible simpler approaches.
- Related to the point above, the paper makes no comparison with previous work, when other heuristics I believe exist for at least some of the questions asked in the paper

---

> ### Author Response · Authors · 2022-08-02
> **Response to reviewer qYgj**
>
> Thanks for your thoughtful and constructive feedback! You asked why we apply neural networks rather than existing algorithms for computing Shapley values. The fastest method to approximate Shapley values (also used in the SHAP package) is a Monte-Carlo approach [1]. A number of other methods exist whose runtime and accuracy depend on the number of samples used, which is usually on the order of several thousand [2, 3, 4, 5]. If the intention is only to calculate Shapley values on a **single** instance, you are indeed right, and the time spent training the network is likely higher than running a Monte-Carlo method on that instance. However, our goal is to compute the solutions for many instances. Once a neural network is trained, computing Shapley values on an instance only requires a couple of matrix multiplications and ReLUs (while the Monte Carlo approach takes thousands of samples, and evaluates the underlying model for each!).
>
> Thus, our innovation lies in speeding up the Shapley/Core computation of **many** instances: we pay an upfront high compute of training a neural network to represent games and compute these solutions (i.e. we distill algorithms for calculating these into a neural network) and obtain a fast constant time (O(1) runtime) per each instance. Based on your comment, we have added a new Appendix where we show that our model-based approach offers an 8x speedup compared to using the SHAP package on a dataset of 9000 samples. By training a neural network on 10% of the data for 30 seconds, we can achieve similar performance while reducing computation time from 4.13 hours to 29 minutes. **Based on your feedback, we have added Appendix I, which captures this point and contains the full experimental details.** The speedup could be significantly higher for much larger datasets, where an accurate Shapley neural model can be trained on a far smaller fraction of the data.
>
> We would like to emphasize that our framework has important motivations aside from allowing for fast measurement of feature importance. A key contribution is evaluating the ability of neural networks to learn how to solve cooperative games. Keep in mind that the solution to a cooperative game is a sophisticated function, mapping from properties of $n$ players (and describing the overall payoffs of $2^n$ coalitions) to $n$ payoffs. Our long-term goal is to create estimators that are scalable, which requires understanding how solutions in small games translate to larger systems. We address this through the novel training procedure of the variable-size models. We show that our models learn how the payoff of every player changes when we introduce a new player, including for games with more players than the model has ever encountered during training.
>
> **You also asked whether neural networks are actually necessary in contrast to linear models or heuristics.** Based on your feedback, we considered a prominent heuristic that divides the payoff in proportion to the weights, that is, $p_j = w_j / (\sum_i^n w_i)$ for each player $j = 1, \dots, n$, and found that our fixed-size models significantly outperform this heuristic. We also observe that linear models do not have the capacity to learn accurate representations of the aforementioned solution concepts. Benchmarking our neural networks with multinomial regression models confirms that our models yield significantly better predictions, outperforming the linear models by 45 to 90 %. **We have added these new results to the new Appendix J and our code base, which we reuploaded.**
>
> Regarding the relative pros and cons of using neural network (black box) models: neural networks can achieve a large speedup over exact or even approximate computations when there are many inputs. While this is a black box tool, the output is interpretable and relates to the most powerful players (WVGs) or key features driving a prediction. As we openly discuss, a drawback of our approach is that our models can lose some accuracy on certain inputs (see Section 4.1.2) and have particular difficulty with capturing discontinuous changes in the solutions. These discontinuities appear across solution concepts because the small set of winning coalitions may drastically change when the quota or weights are slightly altered.
>
> Once again, thank you for taking the time to provide such in-depth feedback. We are happy to answer any additional questions. We also updated our code base to include the code that produced the additional analysis.
>
> ---
>
> [1] Lundberg et al., A Unified Approach to Interpreting Model Predictions (2017)
>
> [2] Leech, D. (2002). Computation of power indices.
>
> [3] Datta et al "Algorithmic transparency via quantitative input influence: Theory and experiments with learning systems." IEEE 2016.
>
> [4] Maleki et al. "Bounding the estimation error of sampling-based Shapley value approximation." arXiv (2013).
>
> [5] Bachrach, Yoram, et al. "Approximating power indices: theoretical and empirical analysis." AAMAS 2010

---

> > ### Comment · Reviewer_qYgj · 2022-08-07
> > **Thanks a lot for the additional results and the thoughtful response!**
> >
> > I think it'd be nice to include the motivation of many vs a single instance in the main body of the paper. I think this really improves the motivation for the authors' current work. I also think it is nice that the authors now provide evidence on how their framework performs compared to at least a simple heuristic, though I think it would be nice to include other sampling/Monte-Carlo based methods found in the literature (even if it is a bit unfair to compare methods that are developed for specific instances to a more general purpose one). I've increased my score to reflect this.

---

> > > ### Author Response · Authors · 2022-08-08
> > > **Thank you for engaging and discussing**
> > >
> > > Thank you so much for your feedback and questions.
> > > > I think it'd be nice to include the motivation of many vs a single instance in the main body of the paper. I think this really improves the motivation for the authors' current work.
> > >
> > > We wholeheartedly agree with you, and if accepted we will use the extra page in the CR version to move the many vs a single instance to the main paper.
> > >
> > > > I also think it is nice that the authors now provide evidence on how their framework performs compared to at least a simple heuristic, though I think it would be nice to include other sampling/Monte-Carlo based methods found in the literature (even if it is a bit unfair to compare methods that are developed for specific instances to a more general purpose one). I've increased my score to reflect this.
> > >
> > > We agree and will update our final version to include results from [1,2] for the feature importance games (Shapley) and [3] for the weighted voting games.
> > >
> > > -----------------------------------------------
> > > [1] A Datta et al., “Algorithmic transparency via quantitative input influence: Theory and experiments with learning systems.” IEEE 2016.
> > > [2] Bachrach, Yoram, et al. "Approximating power indices." Proceedings of the 7th international joint conference on Autonomous agents and multiagent systems-Volume 2. 2008.
> > > [3] Fatima, Shaheen S., Michael Wooldridge, and Nicholas R. Jennings. "A linear approximation method for the Shapley value." Artificial Intelligence 172.14 (2008): 1673-1699.

---

### Author Response · Authors · 2022-08-02
**Summary of our changes**

Dear reviewers and area chair,

We thank all of the reviewers for their insightful comments and feedback, which we believe improved our paper immensely. We have extensively revised the manuscript in accordance with your suggestions. Overall, we made the following changes:

- As inquired by reviewers one and three, we clarified the motivation and experimental procedure behind an important use-case of our framework, which is **speeding up Explainable AI methods of many instances**.
- Addressing reviewer one's remark, we explain why we utilized neural networks and demonstrate that our models outperform a heuristic and multinomial logistic regression models.
- Based on reviewer's two valuable suggestions we:
1. Apply our models to a real-world use case: analyzing voting power in the EU Council.
2. Perform a qualitative analysis that reveal insightful limitations of our models.

Please find full details on our new additions in the responses below and **Appendices I to M**.

We thank the reviewers again for their time and valuable comments.

---

### Meta-Review · Area_Chair_2cDT · 2022-08-23

**Recommendation:** Accept
**Confidence:** Less certain

**Metareview:**

This paper proposes a deep learning approach to computing agent/feature importance in complex cooperative game theoretic settings.  The reviewers are overall positive, albeit some are a bit lukewarm, about the paper.  Overall, the consensus after the discussion is that the idea of studying how well neural computation can be used to approximate the relevant quantities (hard to solve exactly) in cooperative game theory is itself a contribution, and this paper has done a decent job in analyzing the approach and establishing its validity. We would encourage the authors to incorporate the reviewer comments (e.g., move the many vs one discussion to the main paper to strengthen the motivation, include comparisons with more baselines, etc) into the final version and produce a stronger paper.

**Award:**

No

---

### Decision · Program_Chairs · 2022-09-14

Accept